# Oxidized Chitosan-Tobramycin (OCS-TOB) Submicro-Fibers for Biomedical Applications

**DOI:** 10.3390/pharmaceutics14061197

**Published:** 2022-06-03

**Authors:** Zhen Li, Shunqi Mei, Yajie Dong, Fenghua She, Chengpeng Li, Yongzhen Li, Lingxue Kong

**Affiliations:** 1Institute for Frontier Materials, Deakin University, Geelong, VIC 3216, Australia; zkj@deakin.edu.au (Z.L.); dongy@deakin.edu.au (Y.D.); mary.she@deakin.edu.au (F.S.); 2Hubei Digital Textile Equipment Key Laboratory, Wuhan Textile University, Wuhan 430073, China; 3Agricultural Product Processing Research Institute, Chinese Academy of Tropical Agricultural Sciences, Zhanjiang 524001, China; lichengpeng@gdou.edu.cn

**Keywords:** chitosan, tobramycin, centrifugal spinning, submicro-fibers, biomedical application

## Abstract

Chitosan (CS) is a biodegradable, biocompatible, and non-toxic natural amino-poly-saccharide with antibacterial ability, owing to its positively charged amino groups. However, the low charge density leads to poor antibacterial efficiency which cannot meet the biomedical application requirements. In this study, Tobramycin (TOB) was grafted onto the backbone of oxidized chitosan (OCS) to synthesize oxidized chitosan-tobramycin (OCS-TOB). FTIR, ^1^H NMR and elemental analysis results demonstrated that OCS-TOB was successfully synthesized. OCS-TOB/PEO composite fibrous materials were produced by a self-made centrifugal spinning machine. In vitro experiments showed that cells proliferated on the submicro-fibrous OCS-TOB/PEO of appropriate concentration, and the antibacterial ability of OCS-TOB was much improved, compared with pristine CS. The results demonstrated that OCS-TOB/PEO nanofibrous materials could potentially be used for biomedical applications.

## 1. Introduction

As a natural amino-poly-saccharide, chitosan (CS) has been widely used for biomedical applications because it is biodegradable, biocompatible, non-toxic and antibacterial [1]. Its antibacterial ability is mainly attributed to its amino groups located on the backbone of CS. This unique ability is different from other nature biomaterials, such as gelatin [2] and zein [3]. Those positively charged amino groups can directly attract the negatively charged bacterial cell membrane [4], which is different from the antibiotics via a target-specific effect [5]. However, the low charge density of CS limits its antibacterial efficiency [6]. Additionally, natural CS can only be dissolved into an aqueous acidic solutions, which hinders its application [7]. In order to improve the antibacterial ability and the water solubility, therefore, it is important to modify CS backbone by grafting abundant amino and hydroxyl groups.

Tobramycin (TOB) is a water-soluble and broad-spectrum aminoglycoside antibiotic with abundant amino groups and has been widely utilized for anti-infection [8]. Recently, TOB has been loaded in poly(acrylic acid)-grafted-chitosan bio-film to improve the antibacterial activity [9], and loaded in chitosan-coated poly(lactic-co-glycolic acid) (PLGA) nanoparticles to study its effectiveness against chronic *P. aeruginosa* infection for pulmonary treatment [10]. Loading low or non-cytotoxicity drugs onto natural bio-polymers with non-toxic solvent to minimize the side effect of artificial biomaterials is still a challenge.

Compared with hydrogels [11], nanoparticles [12] and nanofilms [13], nano and submicro-fibrous biomaterials have huge potential biomedical applications, owing to their structural features and functions that are similar to the native extracellular matrix (ECM) [14] and conductive to cell adsorption and proliferation [15]. Centrifugal spinning is used to produce uniform nanofibers with significantly high production rate (1 mL per minute for lab scale) and low energy consumption compared with electrospinning [16,17] and blow spinning [18]. Moreover, the uniform nanofibers produced via centrifugal spinning are 3D structures and are highly porous, which will be helpful to improve cell adsorption and proliferation in biomedical applications. The process of centrifugal spinning can be divided into three stages: jet-initiating, jet-extension and solvent evaporation [19,20]. In brief, the liquid polymer material (solution or melt) is ejected through the designed nozzles by centrifugal force, then the jet liquid is extended before the solidified nano-fibers are formed and deposited on the collector after the solvent evaporation or melt cooling [21]. Owing to the large-scale production with low cost, centrifugal spinning has a huge potential for commercial bio-medical applications [14,22,23].

In the current study, we grafted TOB onto oxidized chitosan (OCS). Owing to the abundant amino and hydroxyl groups on TOB, the antibacterial property and water solubility of CS/TOB composites was improved. Furthermore, the toxic side effects and aminoglycoside resistance of TOB could be alleviated or eliminated, since the TOB molecules are fixed on a CS backbone and their motion can be restricted by the macromolecular chain of CS. Meanwhile, polyethylene oxide (PEO), a biodegradable and biocompatible synthetic polymer [24], was mixed with synthesized OCS-TOB solution to improve the spinnability of OCS-TOB. Therefore, uniform OCS-TOB/PEO submicro-fibers were fabricated via a centrifugal spinning method. Chemical structure, thermal property, wettability, mechanical property, antibacterial activities and cytotoxicity of the produced OCS-TOB/PEO submicro-fibers were also discussed and analyzed systematically in this study. It is anticipated that the novel chitosan composite sub-microfibers with strong antibacterial ability and water solubility have great potential for biomedical application.

## 2. Materials and Methods

### 2.1. Materials

Chitosan (86.5% degree of deacetylation, viscosity 100~200 mPa s^−1^) was purchased from Qingdao BZ Oligo biotech CO (Qingdao, China). Tobramycin (TOB) was purchased from Shanghai McLean Biochemical Technology Co., Ltd. (Shanghai, China). Polyethylene oxide (PEO, average Mw~1,000,000) was purchased from Aladdin Reagents Co., Ltd. (Shanghai, China). Escherichia coli (*E. coli*, CMCC(B) 44103) and Staphylococcus aureus (*S. aureus*, ATCC 25923) were obtained from the China Center of Industrial Culture Collection, Beijing, China. All other reagents were analytical grade and used without further purification.

### 2.2. Synthesis of OCS-TOB

The purified CS was oxidized using procedures reported previously with slight changes [25]. CS powders were added into 1% of glacial acetic acid aqueous solution and stirred until they were entirely dissolved. The obtained solution was filtered by microporous membrane filter (Nylon, pore size 1 µm) to remove the tiny undissolved particles and achieve the CS solution. The CS solution was neutralized with diluted NaOH solution (1 × 10^−3^ mol/L) and the purified CS was obtained after drying.

3 g purified CS was dissolved in 200 mL of 1% of glacial acetic acid aqueous solution and stirred 12 h to achieve a homogeneous CS solution. Then, 100 mL sodium periodate (1.99 g) aqueous solution was added into the purified CS solution. The solution was allowed to react in darkness at 40 °C for 1 h. Ethylene glycol (1.74 g) was added to terminate the reaction. The reacted solution was dialyzed in a dialysis bag (8–14 kDa) for three days and lyophilized. OCS was then obtained. In order to get rid of the insoluble gel OCS, which was attributed to the Schiff base reaction, the obtained OCS was dissolved in 1% acetic acid and filtered by a nylon micro-pore membrane.

The purified OCS powders were introduced into 10 *w*/*v*% TOB aqueous solution, and 2 drops of triethylamine (TEA) was added into the mixture solution and reacted on a shaker at 100 rpm for 6 h (pH = 6–7, 55 °C). After reaction, sodium cyanoborohydride solution was introduced for 1 h at 30 °C. Synthesized OCS-TOB was then washed with anhydrous ethanol two to three times so as to remove residual TOB, TEA and NaCNBH_3_. The reacted solution was dialyzed in a dialysis bag (8–14 kDa) for three days and lyophilized. Finally, purified OCS-TOB was obtained.

### 2.3. Fabrication of OCS-TOB/PEO Submicro-Fibers

Predetermined PEO powder was added into DI water to prepare the 7 *w*/*v*% PEO solution. 0.7 g OCS-TOB powders were introduced into 20 mL, 30 mL and 40 mL 7 *w*/*v*% PEO solutions, respectively. Therefore, the mass ratio of OCS-TOB to PEO in three solutions was 1:2, 1:3 and 1:4. OCS-TOB/PEO solutions at these three different ratios were stirred for 8 h at 25 °C so as to obtain homogeneous OCS-TOB/PEO solutions.

In the process of OCS-TOB/PEO submicro-fibers fabrication, the following key operational conditions in centrifugal spinning were used: rotational speed: 4500 rpm; inner diameter of nozzle: 0.31 mm; nozzle length: 5 mm; turning radius of nozzle: 10 cm and collection distance: 30 cm. All experiments were operated at room temperature with 46%RH.

### 2.4. Characterizations

Fourier transform infrared spectroscopy with a universal attenuated total reflection (FTIR-ATR, Vertex-70, Bruker, Karlsruhe, Germany) was used to determine chemical characteristics. All submicro-fiber scaffolds were tested at wavelengths in the range of 4000–400/cm and at a resolution of 1/cm with 64 scans. The ^1^H nuclear magnetic resonance spectrum (^1^H NMR) was tested by a Fourier Nuclear Magnetic Resonance Instrument (400M, Bruker, Karlsruhe, Germany) using a mixed solvent of CD_3_COOD and D_2_O at room temperature. Element contents were tested by an elemental analyzer (Flash EA 1112, Thermo Fisher Scientific, Waltham, MA, USA) and the previous calculation method was followed [26].

The morphology of submicro-fibers was observed by scanning electron microscopy (SEM, JSM-7800F, JEOL, Tokyo, Japan). Before testing, the obtained submicro-fibers were coated with Au by sputter coater (JFC-1600, JEOL, Tokyo, Japan), and the diameters of the fibers were measured by ImageJ software. For each fiber specimen, three different SEM images were used and 100 counts of fibers were measured to calculate fiber diameters.

AR-2000ex rheometer (TA Instruments, New Castle, DE, USA) was used to test the viscosities of polymer solutions at room temperature. A parallel plate (40 mm diameter) was applied to test each sample with a gap in 500 μm. The viscosity of polymer solutions was tested with an increasing shear rate from 0.1/s to 1000/s. The thermal properties of fibers were tested by Thermogravimetric (TGA, 209F1, Netzsch, Selb, Germany). Approximately 5 mg fibers were used for each type of sample. For each testing, the temperature increased from 30 °C to 800 °C with a heating rate of 10 °C/min, and this decreased from 800 °C to 30 °C with a cooling rate of 20 °C/min. A contact angle goniometer equipped with a CCD camera (Kruss, DSA 100, WCA, Frankfurt, Germany) was used to test the contact angles of DI water on the surface of the produced submicro-fiber mats. For each contact angle test, 10 μL DI water droplet was gently placed on the surface of the sample via a micro-plate syringe. The mechanical test of submicro-fibers was performed via an Instron 5943 with a 50 N load cell. The submicro-fibers were twisted into a yarn with 0.5 mm diameter and cut into 1 cm long, and the extension rate was set at 2 mm/min. Young’s modulus and failure work were calculated from the recorded stress-strain curve. At least three tests for each type of sample were undertaken.

### 2.5. Cell Viability Assay

The cell viability assessment of OCS-TOB/PEO submicro-fibers was evaluated via mouse fibroblast cells (L929) using an MTT assay. Briefly, the L929 cells were seeded into 96-well plates with a density of 5 × 10^3^ cells per well and cultured at 37 °C for 24 h in an incubator with a moist air condition of 5% CO_2_. The cultured cells were washed by PBS twice and replaced the original medium with 100 μL sample solution at various concentrations (50–6400 μg/mL). After 24 h incubation, 20 μL MTT (0.5 mg/mL) was added into each well for another 4 h incubation. After removing the mixed medium, 150 μL DMSO was added to dissolve the insoluble formazan crystals. Finally, the dissolved solution was shaken for 10 min and measured by an enzyme-linked immune analyzer at 570 nm. The cells of the control group did not receive further treatment and all experiments were operated six times. The cell viability was expressed as a percentage according to the following formula:(1)$$Cell\;viability\;\left(\%\right)=\frac{OD_{sample}-OD_{blank}}{OD_{control}-OD_{blank}}\times 100\%$$
where, $OD_{sample}$, $OD_{control}$ and $OD_{blank}$ are the optical density of sample, control and blank, respectively.

### 2.6. Antibacterial Assay

An inhibition zone experiment was utilized to assess the antibacterial properties of OCS-TOB via the AGAR disc diffusion method [27]. The antibacterial abilities of OCS-TOB powders were compared with those of CS powders and traditional water-soluble chitosan (WSCS) powders. Furthermore, different ratios of OCS-TOB/PEO submicro-fibers were also used to test their antibacterial abilities with a control. *E. coli* and *S. aureus* were selected to assess the antibacterial properties of prepared materials.

All prepared samples were sterilized by UV for 30 min before experiments. Sterile water and glacial acetic acid were used to prepare 0.5% dilute acetic acid, and then chitosan samples were dissolved into the prepared 0.5% dilute acetic acid to prepare 1 wt.% drug solution. A bacterial solution was incubated with nutrient broth in a shaker at 120 rpm at 37 °C for 18 h. The bacterial solution was diluted 300–400 times. One mL of diluted bacterial solution was dropped into AGAR at 50 °C and then was allowed to cool and coagulate. A 10 μL drug solution was dropped into a 5 mm diameter disc and left for 30 min. The inhibition zone was photographed, and the diameter of the inhibition zone was measured after 24 h culture at 37 °C.

### 2.7. Statistical Analysis

The statistical analysis of all data was calculated as mean + standard deviation (SD) and performed using one-way ANOVA. A probability value *p* < 0.05 was considered statistically significant.

## 3. Results and Discussion

### 3.1. OCS-TOB Syntheses and Characterizations

#### 3.1.1. Mechanism of Synthesis

The synthesis mechanism of OCS-TOB is shown in Figure 1a. First of all, CS is oxidized by NaIO_4_, resulting in the cleavage of C2-C3 bonds and the formation of aldehyde groups. Secondly, TOB is grafted onto the OCS by the Schiff base reaction. Finally, the -C=N- bond between TOB and OCS is replaced by -C-NH-, and TOB was grafted onto the backbone of OCS after the reductive amination by NaCNBH_3_.

#### 3.1.2. FTIR Analysis

FTIR spectra for synthesized OCS-TOB are shown in Figure 1b. For CS, the peak at 3423 cm^−1^ corresponds to the O-H stretching vibration, while the peaks at 2918 cm^−1^ and 2854 cm^−1^ correspond to the C-H stretching vibration; the peak at 1641 cm^−1^ corresponds to the O-H and C=O stretching vibrations; 1550 cm^−1^ corresponds to the N-H bending vibration; 1424 cm^−1^ and 1384 cm^−1^ corresponds to the C-H bending vibration; and 1159 cm^−1^ and 1100 cm^−1^ correspond to the C-O-C stretching vibration. In the OCS curve, 1744 cm^−1^ corresponds to the C=O stretching vibration in aldehyde carbonyl or carboxylic acid carbonyl, while the C-O-C stretching vibration is transferred to 1155 cm^−1^ and 1076 cm^−1^, and the other peaks are consistent with CS. This indicates that CS is oxidized. In the OCS-TOB curve, the peaks at 1571 cm^−1^ and 1550 cm^−1^ correspond to the N-H bending vibration in TOB and OCS, respectively; the peaks at 1155 cm^−1^ and 1077 cm^−1^ correspond to C-O-C stretching vibration. The results demonstrate that CS and TOB share the functional groups (-NH and C-O-C), and the C-O-C stretching vibration in TOB is covered by that in CS, while two N-H bending vibrations indicate that a very small amount of TOB is grafted onto OCS.

#### 3.1.3. NMR Analysis

The ^1^H NMR spectra of CS, OCS and OCS-TOB are shown in Figure 1c. Signal peaks of CS are consistent with a previous study [25]. The peaks at 2.98 ppm and 4.67 ppm are H2 and H2′, respectively; the multiple peaks between the range of 3.50–3.70 ppm correspond to methylene protons H3, H4, H5 and H6. After oxidation, the periodate ion cleaved the C-C bond, the intensity of peak at 2.98 ppm decreases, and a new peak appears at 8.07 ppm in OCS. It indicates that H on C2 decrease and the aldehyde group is generated. In the OCS-TOB spectrum, a new peak at 8.14 ppm appears owing to the protonated hydrogen of C on C=N during Schiff’s base reaction between the aldehyde group of OCS and the terminal amino group of TOB.

#### 3.1.4. Elemental Analysis

The mass percentage of chemical elements can demonstrate their quantitative relationship in a particular compound [25]. The elemental analyses of CS, OCS and OCS-TOB are presented in Table 1. The ratio of C/N in OCS is higher than that in CS, which is attributed to the emission of ammonia during the oxidation of CS [28]. The N percentage in OCS-TOB is lower than that of OCS, owing to the grafting of TOB onto OCS. According to the C/N changes, the grafting rate is approximately 18.3%.

### 3.2. Characterization of OCS-TOB/PEO Submicro-Fibers

#### 3.2.1. Submicro-Fiber Morphology

The morphology of nano-fibers has a significant influence on cell proliferation in biomedical applications [15]. In the centrifugal spinning method, the nano-fiber morphology was determined by the rheological properties of the polymer solution and operational conditions [29,30], including viscosity, rotational speed, nozzle diameter and nozzle length [17].

The centrifugal spinning machine was self-designed and assembled via the “Digital Twins” conception, as shown in Figure 2a,b. It means that a simulated machine is made in a virtual world before manufacturing a real one, then the mechanical structure of the machine can be optimized and improved during simulation so as to save time and cost. Figure 2c shows that the viscosity of the 1:2 OCS-TOB/PEO solution starts from around 59 Pas, peaking to about 63 Pas at 0.2/s, and then decreasing with the increasing shear rate. When the shear rate is higher than 100/s, the relationship between solution viscosity and shear rate becomes almost linear. All OCS-TOB/PEO solutions have similar rheological behaviors. The viscosities for the three OCS-TOB/PEO solutions of different ratios are very similar, although the 1:2 OCS-TOB/PEO solution has a slightly higher viscosity. This indicates that the rheological behaviors of the polymer solution are not significantly influenced by changing the content of OCS-TOB. Therefore, it could be assumed that the sub micro-fibers produced from these three OCS-TOB/PEO solutions of different ratios could have similar morphologies if other operational conditions are remained the same.

The SEM images and fiber diameter distributions of OCS-TOB/PEO sub micro-fibers of the three different ratios are shown in Figure 2d–f. The average diameter of sub micro-fibers increases slightly from 668 nm to 701 nm when the ratio of OCS-TOB/PEO decreases from 1:2 to 1:3. However, the average diameter slightly decreases to 624 nm at 1:4 OCS-TOB/PEO. In addition, the standard deviation of the three samples remains at about 200 nm, regardless of the changing ratio of OCS-TOB/PEO.

#### 3.2.2. Thermal Properties

The thermal properties of the produced OCS-TOB/PEO sub micro-fibers were characterized by thermogravimetric analyses (TGA), and are shown in Figure 3a. The weight of these three samples hardly decreases before 250 °C, which means that the produced sub micro-fibers are highly desiccative [17]. When the temperature reaches 250 °C, the composite sub micro-fibers start degrading. Between 250–350 °C, all samples have an obvious mass loss, and the mass loss increases with the increasing OCS-TOB content. The sample with a higher content of OCS-TOB leads to a bigger mass loss in this range. This might be because the nitrides (such as amino groups) are degraded at this stage. At the range of 350–410 °C, the mass of these samples has declined dramatically, and all of them have less than 20 p of the total mass remaining after 410 °C. The sample with a higher content of PEO leads to a bigger mass loss at this stage. This might attribute to carbides (including aldehyde groups) that are carbonized. After that, the samples are stable and hardly lose any mass, and the sample with higher percentage of OCS-TOB has a higher residual mass.

#### 3.2.3. Mechanical Properties

The mechanical properties of OCS-TOB/PEOs with different ratios are shown in Figure 3b. For 1:2 OCS-TOB/PEO, the tensile strain is slightly higher than 100% and the tensile stress is 4.2 MPa; for 1:3 OCS-TOB/PEO, the tensile strain and tensile stress increase to around 150% and 6.2 MPa, respectively; and for 1:4 OCS-TOB/PEO, the tensile strain and tensile stress further increase to approximately 180% and 8.2 MPa, respectively. The result indicates that when the content of OCS-TOB decreases from 1:2 to 1:4, both of the tensile strain and tensile stress of OCS-TOB composite sub micro-fibers are significantly increased. Therefore, the mechanical strength of OCS-TOB/PEO composite sub micro-fibers increases with increasing the percentage of PEO. All three OCS-TOB/PEO sub micro-fibers are strong enough for cell migration and proliferation in tissue engineering applications as the tensile stress of ECM is at the recommended range of 0.8–18 MPa [31,32].

Mechanical properties of sub micro-fibers are strongly influenced by many factors, including chemical structure of polymers and physical properties of fibers (structure, morphology and diameter). Therefore, the mechanical properties might attribute to the strong intramolecular or intermolecular hydrogen bonds in PEO as reflected by the semi-crystalline phase of PEO [33,34]. Hydrogen bonds also exist in OCS-TOB but are formed between aldehyde group of oxide CS and terminal amino groups [25], which has a relatively lower strength than the hydrogen bonds in PEO. The molecular weight also only has limited effects but cannot be excluded (the molecular weight of PEO is larger than OCS-TOB) [33].

#### 3.2.4. Wettability Studies

Figure 3c–e show the hydrophilicity of the produced OCS-TOB/PEO composite sub micro-fiber mats. The water contact angle of 1:2 OCS-TOB/PEO sub micro-fibers is 83.5°. As the percentage of PEO in OCS-TOB/PEO composite sub micro-fiber scaffold increases, the contact angle orderly decreases to 64.1° for 1:3 OCS-TOB/PEO and 43.3° for 1:4 OCS-TOB/PEO sub micro-fiber mats, respectively. As previously reported, the contact angle of pure PEO cast film is 21° [35]. It demonstrates that the wettability of OCS-TOB/PEO sub micro-fiber mat increases as the content of PEO increases, which might be attributed to the water-soluble ability of PEO which is better than OCS-TOB.

### 3.3. Cell Viability Assay

The viability of L929 on OCS-TOB/PEO sub micro-fibers with different concentrations of medium extracts is shown in Figure 4a. It is observed that the cell viability of 1:4 OCS-TOB/PEO is higher than that of 1:2 OCS-TOB/PEO at the same concentration. Therefore, a higher content of OCS-TOB inhibits cell proliferation, owing to its cytotoxicity. Additionally, the samples at 50 μg/mL and 100 μg/mL showed very similar and high proliferation ability. Then, the cell viability continuously decreases as the concentration increases from 100 μg/mL to 6400 μg/mL. It indicates that a low concentration (50 μg/mL and 100 μg/mL) of OCS-TOB/PEO contributes to cell proliferation, while a high concentration (>100 μg/mL) shows toxicity and significantly decreases the cell viability. The results demonstrate that an appropriate (such as 50–100 μg/mL) concentration of OCS-TOB/PEO can be potentially used for cell proliferation, and lower concentration of drugs usually has a lower drug-induced risk.

### 3.4. Antibacterial Assay

Figure 4b,c and Table 2 show the inhibition effect of chitosan and its derivatives on *E. coli* and *S. aureus*. OCS-TOB has the largest inhibition zones (over 18 mm) in *E. coli* and *S. aureus*, natural CS has limited antibacterial effects on both *E. coli* and *S. aureus*, and traditional water-soluble chitosan (WSCS) hardly inhibits the growth of those two kinds of bacteria. The results demonstrate that WSCS hardly has antibacterial ability, while synthesized water-soluble OCS-TOB significantly improves the antibacterial properties of natural CS. This might be because the positive charge density of OCS-TOB is much higher than the others and therefore, there are more electrostatic attractions between positively charged OCS-TOB and negatively charged bacterial cell membranes, owing to grafted TOB [36].

OCS-TOB is a water-soluble chitosan derivative. OCS-TOB/PEO composite sub micro-fibers of different ratios can inhibit bacterial growth, as shown in Figure 4d,e and Table 2. OCS-TOB/PEO composite sub micro-fibers have a similar effect on inhibitions of both *E. coli* and *S. aureus*, and the size of inhibition zone increases as the OCS-TOB content increasing. However, when the ratio of OCS-TOB/PEO increased to 1:2, the inhibition zone size is still 18.21 mm, which is almost the same as that for 1:3 OCS-TOB/PEO. 1:2 OCS-TOB/PEO does not show larger inhibition zone size than 1:3 OCS-TOB/PEO, which might be because the drug diffusion ability limited inhibition effect. It indicates that 1:3 OCS-TOB/PEO could maximize the antibacterial ability, when considering the cytotoxicity of OCS-TOB.

## 4. Conclusions

In this study, TOB was grafted onto the oxidized chitosan, aiming at improving the antibacterial property and water-solubility of chitosan. The water-solubility of synthesized OCS-TOB was significantly improved to 7 *w*/*v*%, which indicated that TOB was successfully grafted onto the oxidized chitosan. The uniform OCS-TOB/PEO composite sub micro-fiber mats with controlled hydrophilicity properties can be produced by controlling the parameters. Produced OCS-TOB/PEO sub micro-fibers were characterized by FTIR, TGA, and mechanical test. They demonstrated that OCS-TOB/PEO sub micro-fibers via centrifugal spinning method have stable chemical and physical properties. Inhibition zone test demonstrated that the antibacterial properties of OCS-TOB significantly improved (around 19 mm), comparing with natural chitosan (around 9 mm) and WSCS (almost 0 mm). Higher contents of OCS-TOB in OCS-TOB/PEO composite sub micro-fibers lead to better antibacterial effect, however, when the ratio of OCS-TOB/PEO is over 1:3, the antibacterial efficiency of OCS-TOB/PEO composite sub micro-fibers hardly increased. Low concentration (50 μg/mL and 100 μg/mL) of OCS-TOB/PEO sub micro-fibers are conductive to cell proliferation, comparing with high concentration (>100 μg/mL). Overall consideration of both cytotoxicity and antibacterial ability, 1:3 OCS-TOB/PEO composite sub micro-fibrous had shown the potential for medical applications.

## Figures and Tables

**Figure 1 pharmaceutics-14-01197-f001:**
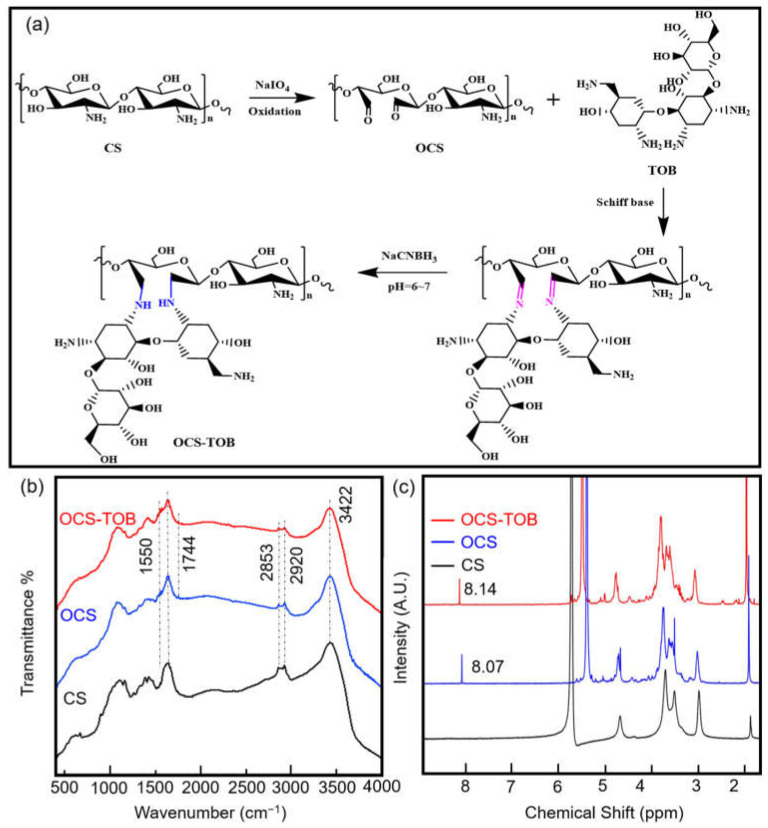
Synthesis of OCS-TOB procedures (**a**); FTIR curves (**b**) and ^1^H NMR spectrum (**c**) of CS, OCS and OCS-TOB.

**Figure 2 pharmaceutics-14-01197-f002:**
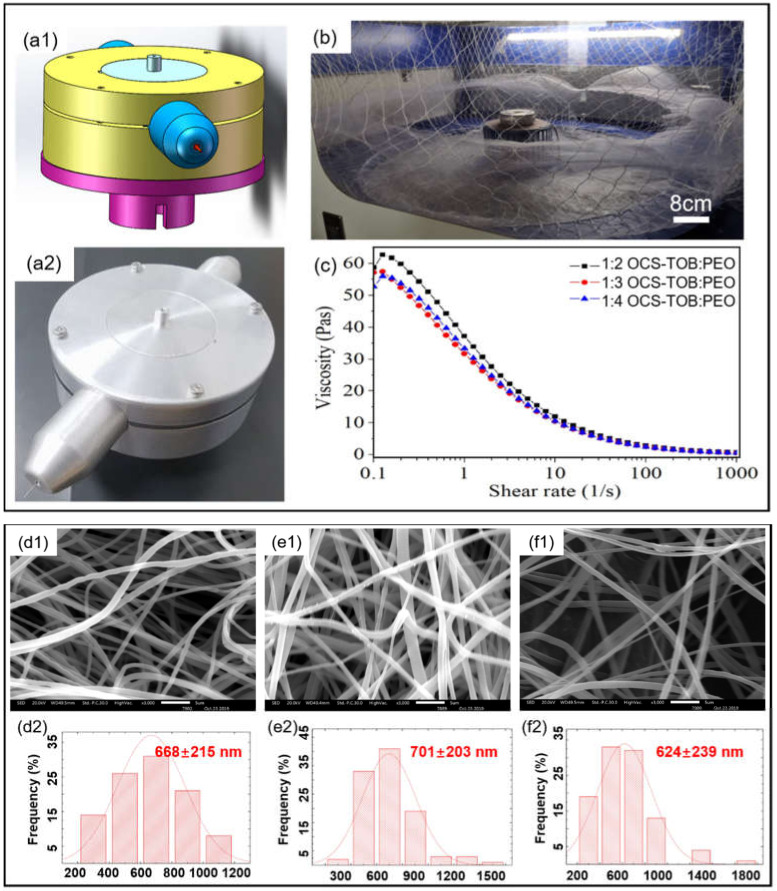
Virtual model and the corresponding optical photo of self-designed centrifugal spinning equipment (**a1**,**a2**,**b**); the viscosities of three different ratios of OCS-TOB/PEO solutions change with shear rate (**c**); SEM images of OCS-TOB/PEO sub micro-fibers with different ratios of solutions with corresponding fiber diameter distribution, 1:2 OCS-TOB/PEO (**d1**,**d2**), 1:3 OCS-TOB/PEO (**e1**,**e2**) and 1:4 OCS-TOB/PEO (**f1**,**f2**), the scale bar is 5 μm.

**Figure 3 pharmaceutics-14-01197-f003:**
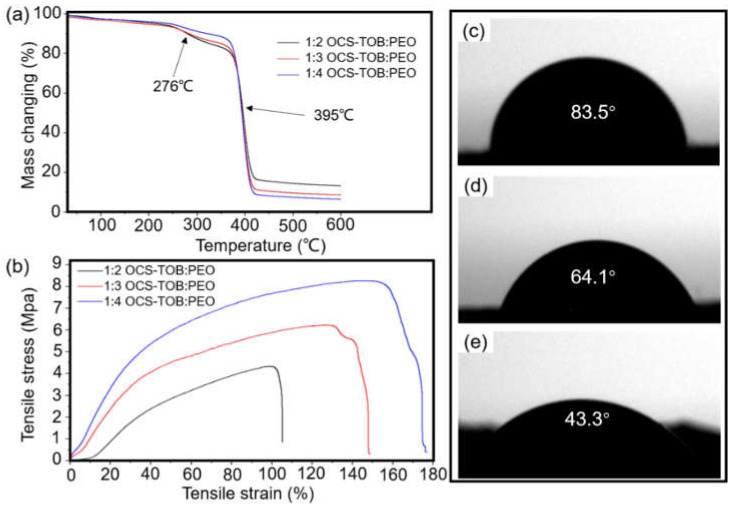
The TGA results of different ratios of OCS-TOB/PEO composite sub micro-fibers (**a**); mechanical test of the produced OCS-TOB/PEO sub micro-fiber yarn (**b**); the contact angle test of different ratios of OCS-TOB/PEO sub micro-fiber mats: 1:2 OCS-TOB/PEO (**c**), 1:3 OCS-TOB/PEO (**d**) and 1:4 OCS-TOB/PEO (**e**).

**Figure 4 pharmaceutics-14-01197-f004:**
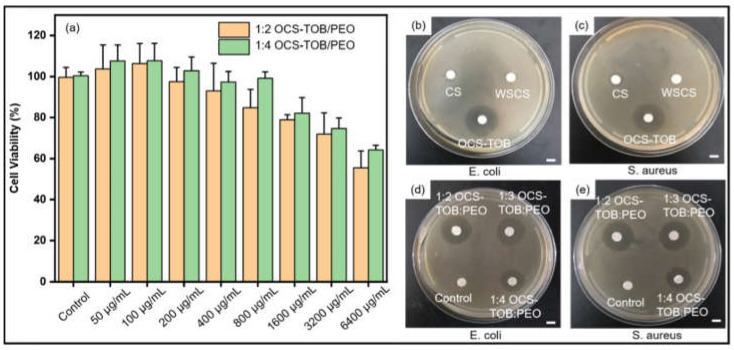
Cell viability of 1:2 OCS-TOB/PEO and 1:4 OCS-TOB/PEO in 24 h (**a**); inhibition effect of CS, WSCS and OCS-TOB against *E. coli* (**b**) and *S. aureus* (**c**); and inhibition effect of different ratios of OCS-TOB/PEO sub micro-fibers against *E. coli* (**d**) and *S. aureus* (**e**). Scale bar is 5 mm.

**Table 1 pharmaceutics-14-01197-t001:** Mass percentage of C and N in CS, OCS and OCS-TOB.

Samples	C (%)	N (%)	C/N
CS	41.8	7.59	5.51
OCS	36.23	6.48	5.59
OCS-TOB	37.90	5.52	6.87

**Table 2 pharmaceutics-14-01197-t002:** Average inhibition sizes of CS, WSCS, OCS-TOB and different ratios of OCS-TOB/PEO sub micro-fibers against *E. coli* and *S. aureus*.

Drug	Average Diameter of Inhibition Zone (mm)		Standard Deviation	
	*E. coli*	*S. aureus*	*E. coli*	*S. aureus*
CS	9.55	8.91	0.31	0.36
WSCS	-	-	-	-
OCS-TOB	19.49	18.66	0.26	0.28
1:2 OCS-TOB/PEO	18.21	18.38	0.21	0.25
1:3 OCS-TOB/PEO	18.17	18.07	0.22	0.23
1:4 OCS-TOB/PEO	15.58	15.67	0.28	0.25
Control	-	-	-	-

## Data Availability

Not applicable.

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
