# Peer review of "Oxidized Chitosan-Tobramycin (OCS-TOB) Submicro-Fibers for Biomedical Applications"

_pharmaceutics, 2022, doi:10.3390/pharmaceutics14061197_

Round 1

Reviewer 1 Report

The manuscript describes preparation of new materials based on oxidized chitosan/tobramycin conjugate. Cytotoxicity and antibacterial properties of the materials were studied. The materials could potentially be used for wound healing. The results are interesting and the manuscript can be accepted after following issues be addressed.

  1. Lines 40-42. "Loading low or non-cytotoxicity drugs onto natural bio-polymer with non-toxic solvent to minimize the side effect of wound dressing biomaterials is still a challenge." Does it mean that toxic solvents were used in [7, 8]?
  2. Line 80. "CS powders were added into glacial acetic acid (1%)". It is not clear what the authors mean here (1% of acid per CS, 1% solution of acid in water and so on). 
  3. Line 86. "Then, 100 mL sodium periodate (1.99 g) was added". What was the solvent? Water? This question concerns all Part 2.2.
  4. Fig. 1. Tobramycin has 4 aminogroups. It is not clear why the authors believe that just these two aminogroups are involved in the interaction with aldehyde groups.
  5. Part 3.3. Why the data for 1:3 OCS-TOB/PEO are not presented?
  6. Lines 304, 305. "Additionally, the cells proliferate at 50 μg/mL with the proliferation ability peaked at 100 μg/mL." It is not clear if these differences are statistically significant.

Author Response

Dear reviewer,

I would like to thank you for providing very constructive comments on our article, which certainly help us improve the quality of the article. The modifications of the article based on your report are listed in the upload file.

Reviewer 2 Report

The paper could be published after revision.

Here are some observations.

The wound healing capability is not proved for the present formulation. A test on mice is recommended as it could be a strong point for the paper. If this is not possible, the authors must remove the wound healing statements from the manuscript and title.
In figure 1a TOB should be placed on the reaction arrow.
Name nano-fibers is incorrect used as by definition the nanomaterials are considered materials of which at least one dimension is between 1 and 100 nm. For the present case, fibers of submicronic diameter describe better the results.
As it plays a crucial role in the bioactivity of the system, cell viability and diameter of inhibition zone of bare TOB should be included for comparison and discussion.
Although the paper is well written, in some parts the text needs to be revised to improve English.

Author Response

(The authors gave the same response as above.)

Reviewer 3 Report

The manuscript entitled ''Oxidized chitosan-tobramycin (OCS-TOB) nanofibers for wound healing'' contains some interesting findings, and it may ultimately be suitable for publication. The authors prepared tobramycin (TOB) covalently attached oxidized Chitosan (OCS-TOB) material and the obtained material was applied for the centrifugal spinning (force spinning) technique to manufacture OCS-TOB nanofibers. Then, the fibers were characterized. Also, the authors measured the antibacterial activity and cell viability of the fibers. However, the manuscript needs many more characterizations and data to improve the results and discussion.  I thus recommend the paper be reconsidered after major revisions.

The reviewer has the following comments  

  1. The title of the manuscript should be modified. Specifically, remove would healing from the title because the authors did not perform any real would healing application, i.e. animal model, and others.
  2. FTIR and NMR spectra of TOB should be included in Figures 1b, and c to understand the final chemical structure of OCS-TOB.
  3. More preparation details of fibers and their spinning conditions, including humidity, exact temperature, etc., should be reported.
  4. If centrifugal spinning can be useful to produce nanofibers from solids/pellet polymers, why did the authors use the OCS-TOB solution to manufacture the fibers? Why didn’t they produce CS-TOB fibers directly from OCS-TOB solid/pellets?
  5. Did the authors crosslink the OCS-TOB fibers? Because without the crosslinking the fibers immediately dissolved in the aqueous solution/buffer. Therefore, it is difficult to measure cell viability and antibacterial studies.  
  6.  DSC analysis, degradation, porosity, and zeta potential of fibers should be reported
  7. The morphologies (SEM) of OCS-TOB nanofibers before and after the antibacterial study should be reported.
  8. The introduction section is very short and should be revised entirely so that the reader can clearly identify the scientific problems solved by this research. There are several biocompatible systems with a wide range of biomaterials, including hydrogels, nanoparticles, nanofilms, nanocomposites, etc., but why did the authors select only nanofibers? The authors should emphasize why the nanofibers are familiar, or favorable compared to other systems using the following articles. Moreover, the information on biomaterials (Chitosan, Gelatin, Zein, and PCL) should be expanded in the introduction. Thus, the following articles should be quoted in the introduction.

Chitosan (https://doi.org/10.1016/j.colsurfb.2021.111819), BSA (https://doi.org/10.1016/j.msec.2020.111698),

Gelatin; (https://doi.org/10.3390/ph14040291, https://doi.org/10.1016/j.jmbbm.2020.103696)

Zein (https://doi.org/10.3390/pharmaceutics11120621).

https://doi.org/10.1016/j.msec.2020.110928. https://doi.org/10.1039/D1EN00354B, https://doi.org/10.3390/pharmaceutics12121208

https://doi.org/10.1016/j.jmbbm.2021.104554

It would be more realistic to cover such kind of research work in the current manuscript. Which will enrich the quality of the current manuscript as well as inquisitiveness to the readers.

  1.  According to the revised data, the conclusions should be modified with more quantitative data. 

Author Response

(The authors gave the same response as above.)

Round 2

Reviewer 3 Report

From the title, and abstract, I had the impression that the authors were very broad and want to cover all types of nanofibers for a broad range of biomaterials.

However, from the chosen literature and statements made in the introduction, it appears that the authors are actually limiting themselves (mostly) to nanofibers that are extruded, forming non-woven mats for the most part. This restriction to a subset of manufacturing techniques seems reasonable to me, as the field of submicron/micro/nanofibers, especially based on gelatin, zein, PCL, PLA, and UHWPE are extremely vast and would likely require a review of its own. If the authors really want to include centrifugal/force spinning submicron/micro/nanofibers in this article, the most important works by Mamidi, EV Barrera, Alex Elias Zuniga, and others should be cited in the introduction.

It would be more realistic to cover such kind of research work in the current manuscript. Which will enrich the quality of the current manuscript as well as inquisitiveness to the readers.

Author Response

Dear reviewer,

Thank you for your further comments and suggestions, which certainly help us to improve the quality of this paper. According to your comments, we studied the research report from Mamidi, EV Barrera and Zuniga, which inspired us a lot. The detail of the improvement has been attached.
